# “Pesto” Mutation: Phenotypic and Genotypic Characteristics of Eight GCK/MODY Ligurian Patients

**DOI:** 10.3390/ijms24044034

**Published:** 2023-02-17

**Authors:** Alessandro Salina, Marta Bassi, Concetta Aloi, Marina Francesca Strati, Renata Bocciardi, Giuseppe d’Annunzio, Mohamad Maghnie, Nicola Minuto

**Affiliations:** 1LABSIEM (Laboratory for the Study of Inborn Errors of Metabolism), Pediatric Clinic, IRCCS Istituto Giannina Gaslini, 16147 Genoa, Italy; 2Department of Neuroscience, Rehabilitation, Ophthalmology, Genetics, Maternal and Child Health (DINOGMI), University of Genoa, 16100 Genoa, Italy; 3Department of Pediatrics, IRCCS Istituto Giannina Gaslini, 16147 Genoa, Italy; 4UOC Genetica Medica, IRCCS Istituto Giannina Gaslini, 16147 Genoa, Italy

**Keywords:** monogenic diabetes, GCK/MODY, glucokinase (GCK), hyperglycaemia

## Abstract

Maturity Onset Diabetes of the Young (MODY) is a monogenic form of diabetes mellitus (DM) that accounts for around 2–5% of all types of diabetes. Autosomal dominant inheritance in pathogenic variations of 14 genes related to β-cell functions can lead to monogenic types of diabetes. In Italy, GCK/MODY is the most frequent form and it is caused by mutations of the glucokinase (GCK). Patients with GCK/MODY usually have stable mild fasting hyperglycaemia with mildly elevated HbA1c levels and rarely need pharmacological treatment. Molecular analysis of the GCK coding exons was carried out by Sanger sequencing in eight Italian patients. All the probands were found to be heterozygous carriers of a pathogenic gross insertion/deletion c.1279_1358delinsTTACA; p.Ser426_Ala454delinsLeuGln. It was previously described for the first time by our group in a large cohort of Italian GCK/MODY patients. The higher levels of HbA1c (6.57% vs. 6.1%), and the higher percentage of patients requiring insulin therapy (25% vs. 2%) compared to the previously studied Italian patients with GCK/MODY, suggest that the mutation discovered could be responsible for a clinically worse form of GCK/MODY. Moreover, as all the patients carrying this variant share an origin from the same geographic area (Liguria), we postulate a possible founder effect and we propose to name it the “pesto” mutation.

## 1. Introduction

Monogenic diabetes is the second most prevailing type of diabetes after type 1 diabetes (T1D) in Italian youth, and the correct aetiological diagnosis greatly affects the treatment and likely the prognosis of diabetes complications [1].

Maturity Onset Diabetes of the Young (MODY) (MIM# 606391) is a monogenic form of diabetes mellitus (DM) that accounts for around 2–5% of all types of diabetes [2,3].

GCK/MODY patients usually have stable mild fasting hyperglycaemia from birth with mildly elevated HbA1c levels (5.6–7.5%) [4], rarely suffer from chronic complications, and, in most cases, no treatment is usually necessary. Insulin therapy and/or oral hypoglycaemic agents are not usually required except during pregnancy, where insulin administration is useful to prevent foetal macrosomia in cases where a foetus has not inherited a maternal GCK disease-causing variant [5]. A study related to Japanese GCK/MODY patients reported that in 9 out of 55 subjects, pharmacological treatment was required [6]. This can be explained by the fact that some of the Japanese patients followed a diet rich in carbohydrates and had sedentary lifestyles.

Currently, autosomal dominant inherited pathogenic variants located in 14 different genes that play a crucial role in in β-cell functions are known to be linked to 14 subtypes of the MODY phenotype [7]. GCK/MODY is the most common form of monogenic diabetes in Italy and the South of Europe [8]. Heterozygous glucokinase gene (GCK, MIM# 125851) pathogenic variants located in the promotor or in the 10 exons and their flanking regions involved in the splicing process are the cause of the disease. The GCK gene maps on chromosome 7p13 and encodes for an enzyme of 465 amino acids that plays a pivotal role in insulin secretion in response to blood glucose levels. In the last 10 years, several large cohorts of patients with the GCK/MODY phenotype have been described [9,10,11], and, to the best of our knowledge, 841 GCK pathogenic variants are present in the HGMD free public database (http://www.hgmd.cf.ac.uk/ac/index.php, accessed on 1 December 2022). Point mutations represent the most common type of pathogenic mutation (N = 598), while gross insertions or deletions and complex rearrangement account for 3.3% of the genetic defects. Recently, Ghadir et al. reported that GCK frameshift mutations are usually associated with elevated blood glucose levels when compared to the missense variants, and insulin treatment may be required in order to maintain good glycaemic control [12].

The research on PubMed of the “founder effect of MODY mutation” allows us to detect 18 different results. Only a few manuscripts in particular aimed to describe the founder effect of GCK mutations in the Southern Italian population [13] and in Central Europe [14,15]. In our previous study [16] on GCK/MODY in the Italian population, we described the clinical and genetic features of two unrelated probands with *GCK* in/del c.1279_1358delinsTTACA, p.(Val427_Ser453delinsLeuGln) (NM_000162.5, NP_000153.1). In the following years, several *GCK* molecular investigations were performed in our centre and the gross in/del c.1279_1358delinsTTACA has been detected in another four unrelated families. In this brief report, our aim is to describe the clinical manifestations caused by the *GCK* pathogenic insertion/deletion c.1279_1358delinsTTACA, p. (Val427_Ser453delinsLeuGln) in a small cohort of eight subjects from six unrelated families from the same geographic area (Liguria), postulating a possible founder effect.

## 2. Results

Anamnestic, clinical, and metabolic data of the eight subjects (three males and five females) are summarised in Table 1. All patients were unrelated except P5-P6, who were uncle and niece, and P7-P8, who were siblings, and all of them were found to be carriers of a pathogenic gross insertion/deletion c.1279_1358delinsTTACA; p. (Val427_Ser453delinsLeuGln) in a heterozygous state (Figure 1a,b). 

All subjects had a normal or low birth weight. The average age at the first finding of hyperglycaemia was 7.9 ± 6.8 years (6 months–20 years). The HbA1c detected at onset was found to be on average 6.57 ± 0.35% (6.2–7.2%) and OGTT after 2 h detected average values of 182.65 ± 37.8 mg/dL. There was absence of β-cell IAA, IA-2A, and GADA in all cases. After the detection of hyperglycaemia, insulin therapy was needed in two cases.

This mutation was reported in our previous manuscript, where a large group of Italian GCK/MODY patients was described [13], and linked for the first time to the GCK/MODY2 phenotype. It was also included in the dbSNP database as rs193922274 without any frequency, and it was absent from Exac, 1000G, gnomAD Exomes, and gnomAD Genomes. ClinVar and ACMG classified it as a “likely pathogenic allele” and “pathogenic”, respectively.

## 3. Discussion

Since 2008, in LABSIEM of IRCCS G. Gaslini, Genoa Italy, 278 unrelated families with the GCK/MODY clinical phenotype were genotyped and found to be carriers of a GCK defect; the insertion/deletion c.1279_1358delinsTTACA in GCK exon 10 was detected in eight patients. This is the most frequent GCK pathogenic variation that we found in our series, accounting for 2.9% of all cases. The deletion of 79 nucleotides and the insertion of TTACA between nucleotides 1279 and 1358 (c.1279_1358delinsTTACA) induce a severe alteration in the translation of the protein, with the loss of part of the “large domain”, part of the “small domain”, and the entirety of “loop 3”. Consequently, we assume that the protein is totally inactive. This type of pathogenic variant is not so rare in *GCK*; up to now, a total of 28 (3.6%) different gross insertions, gross deletions, and complex rearrangements linked to GCK/MODY have been reported in the free public HGMD (accessed on 1 December 2022, https://www.hgmd.cf.ac.uk/ac/index.php). 

The phenomenon of genetic alteration that is observed with high frequency in a group that is geographically or culturally isolated, in which one or more ancestors were carriers of the altered gene, is often called the "founder effect". Studies regarding the founder effect of pathogenic variants, in most of the cases, described groups or cohorts of people that were members of geographically isolated populations or lived in closed communities due to religious or ethnical rules. To the best of our knowledge, only a few groups of reports hypothesised the possibility of a founder *GCK* mutation [13,14,15] or the description of a small group of MODY patients carrying the same mutations originating from the same geographic region: in European MODY populations, five unrelated families from Oxford with p.Gly299Arg [16] and three different mutations in Spain (p.Val182Leu in three families from Cantabria, p.Glu399X in two families from Albacete, and p.Ala379Val in two families from Basquete) [17] were found. In Norway (p.Glu339Gly) [18] and Slovakia (c.-71G>C) [11], ancestral variants have been detected in six and seven families, respectively. The founder effect was also demonstrated in five families with p.Val226Met from Quebec [19]. None of these reports is strictly related to a close community.

In our study, all eight cases came from the urban and suburban area of Genoa, the capital city of Liguria, so we suppose that this gross insertion/deletion may have ancestral origins in this geographical area. For this reason, we would like to propose to name this variant the “pesto” mutation, according to the name of the renowned traditional Genoese sauce.

The GCK/MODY phenotype is highly variable and related to the type of *GCK* pathogenic variation. However, establishing genotype/phenotype correlations is difficult due to the fact that different families with the same defect show different clinical manifestations [8,20,21,22,23,24]. This can be explained by the fact that different life habits, such as diet and sedentary lifestyles, may have a role in the GCK/MODY clinical manifestations. Ghadir et al. reported that *GCK* frameshift mutations are usually linked to elevated blood glucose levels as opposed to the missense ones. The same study also highlights that GCK/MODY patients who were over 40 had significantly higher fasting glucose levels and higher HbA1c levels than subjects under 40 [12]. In our report, mean fasting glucose levels at onset were over 122 ± 15.8 mg/dL, with mean Hb1Ac levels of 6.57 ± 0.35%. GCK/MODY patients generally present at onset with glycaemic values below 124 mg/dL (impaired fasting glucose) and therefore pharmacological treatment is not required. We previously reported [20] that only two probands, both un-carriers of the “pesto” mutation, out of 98 GCK/MODY cases (2%) were treated with insulin one year after the first finding of hyperglycaemia. In order to prevent hyperglycaemic events, insulin treatment was required in two subjects in our cohort of “pesto” mutation patients, P1 and P5 (25%). The higher levels of HbA1c (6.57% vs. 6.1%), and the higher percentage of patients requiring insulin therapy (25% vs. 2%) compared to the previously studied Italian patients with GCK/MODY, suggest that the “pesto” mutation could be responsible for a clinically worse form of GCK/MODY.

In particular, we wish to underline that in a female patient (P4), elevated hyperglycaemia (151 mg/dL, see Table 1) in the absence of β-cell autoantibodies was measured when she was 6 months old. Clinicians firstly suspected a neonatal diabetes mellitus diagnosis, due to the maternal family history of diabetes mellitus. After genetic counselling, the diabetologist chose to request *GCK* sequencing in the proband and her parents. The gross indel was therefore found in P4 and her mother; it was undetected in all healthy family members that gave consent for genetic testing. In conclusion, GCK/MODY was diagnosed. Based on our experience, we suggest to consider GCK/MODY when probands with a positive history of diabetes mellitus manifest hyperglycaemia onset before 1 year of age. 

## 4. Materials and Methods

All patients were selected by applying criteria established by the ISPAD Clinical Practice Consensus Guidelines for monogenic diabetes, which suggest the targeted genetic analysis of patients with a clinical diagnosis of MODY [2,25,26]. These criteria include age at hyperglycaemia onset <25 years, fasting glucose level >99 mg/dL, OGTT after 120 min > 140 mg/dL, absence of β-cell autoantibodies, family history of diabetes mellitus (DM), absence of obesity.

All the subjects included were originally from the Liguria region (Northwest of Italy). Anamnestic, metabolic, and clinical data of 8 probands belonging to 6 unrelated families and their relatives were collected. DNA was extracted from a whole blood sample using a QIAMP DNA Blood MIDI Kit (QIAGEN, USA). Molecular analysis of the 10 GCK coding exons and their flanking regions was performed by direct Sanger sequencing as previously described, alongside the procedure applied to define the identified del/ins [16]. The sequence of oligonucleotides used in this work for GCK analysis is available upon request. Amplicons were purified by the Exonuclease I/Shrimp Alkaline Phosphatase Method (USB, Douglasville, GA, USA) and directly sequenced in both directions using the Big Dye Terminator v1.1 sequencing kit (Applied Biosystems, Waltham, MA, USA) and ABI PRISM sequencing apparatus 3730 (Applied Biosystems, USA). The generated sequences were aligned with reference sequence NM_000162.3 and analysed with Sequencer 5.0 software (Gene Code Corporation, Ann Arbour, MI, USA). All the variations detected were validated by sequencing both DNA strands of two independent PCR products. The role of pathogenicity of the identified variant was evaluated using ClinVar (accessed on 1 December 2022, https://www.ncbi.nlm.nih.gov/clinvar/) and Varsome (accessed on 1 December 2022, https://varsome.com/) [27,28,29].

After genetic counselling, informed written consent for genetic testing was obtained from all participants. A questionnaire for clinical evaluation and enrolment was submitted to all participants. The research was conducted according to the Helsinki Declaration and approved by the institutional ethics committee. 

## 5. Conclusions

In humans, it is known that cultural and/or geographical isolation promotes the formation of closed communities where the recruitment of newcomers is difficult and people tend to marry within the community. This allows the insurgence of pathogenic mutations in this community, creating serious health problems. Our report is the second Italian study that aimed to describe the ancestral origin of GCK pathogenic variants in six non-consanguineous families that live in the area of Genoa. The moderately high number of cases collected (N = 8) allowed our multidisciplinary group, composed of a biologist, paediatric diabetologist, and endocrinologist, to better understand the clinical implications and to describe the molecular effects on enzyme activity provoked by this rare insertion/deletion.

In the near future, with the enrolment of new paediatric patients with clinical manifestations of hyperglycaemia or diabetes mellitus at the Regional Center of Diabetologia, IRCCS G. Gaslini, we will aim to perform molecular investigations through Sanger or next-generation sequencing (targeted panel or whole-exome sequencing) in order to confirm the presence of insertion/deletion c.1279_1358delinsTTACA in Ligurian paediatric patients and young adults. This approach will allow our group to detect other pathogenic variants causative of monogenic diabetes mellitus and, in case of their absence, discover new genes that play a pivotal role in the release of insulin in response to blood glucose levels. A diagnosis of MODY in childhood is important to avoid the consequences of chronic hyperglycaemia, the use of incongruous therapies, and, given the autosomal dominant inheritance, to provide genetic advice.

In conclusion, Liguria is not a culturally and geographically closed region. The presence of the insertion/deletion c.1279_1358delinsTTACA; P. (Val427_Ser453delinsLeuGln), in all eight Ligurian subjects, leads us to suppose that the “pesto mutation” may have ancestral origins in Liguria and so reflect a founder effect. “Pesto” mutation patients have a more severe clinical GCK/MODY phenotype and they often require pharmacological treatment.

## Figures and Tables

**Figure 1 ijms-24-04034-f001:**
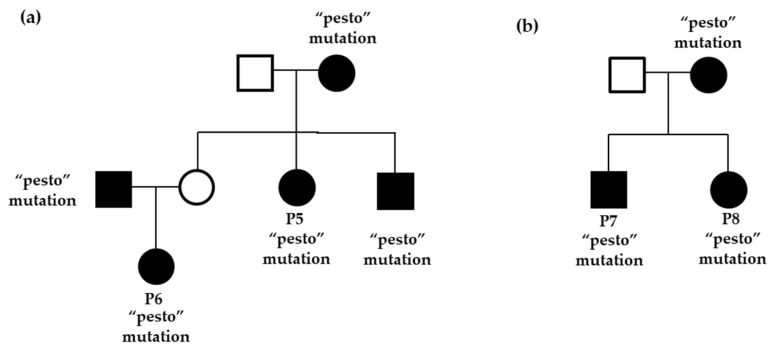
Pedigree of the “pesto” mutation (c.1279_1358delinsTTACA, p. (Val427_Ser453delinsLeuGln)) found in *GCK* in related patients: (**a**) P5-P6 (uncle and niece) and (**b**) P7-P8 (siblings).

**Table 1 ijms-24-04034-t001:** Clinical features and age at diagnosis in eight c.1279_1358delinsTTACA *GCK* mutated patients.

P	Sex (M/F)	Birth Weight (Grams)	E.G (Weeks)	Age at Hyperglycaemia Diagnosis (Years)	1st Fasting Glucose Level (mg/dL)	BMI (kg/m^2^)	OGTT 120 min (mg/dL)	HbA1c (%)	GDM	Affected Family		Pharmacological Treatment (Within 1 Year of Diagnosis)
										Member	HbA1c (%)	
1	M	2700	38	15	131	20	169	6.7	no	father	6.5	insulin
2	M	3250	40	4	120	15	160	6.4	no	mother	nd	no
3	M	2900	41 + 2	3	115	14.1	nd	6.2	no	father	6.7	no
4	F	1665	31 + 6	6 months	151	14.6	nd	6.2	yes	mother	5.4	no
5 *	F	3100	40	20	130	18	nd	7.2	yes	mother	nd	insulin
6	F	2680	40	3	96	14.4	171	6,5	no	father	6.5	no
7	M	2500	37	7	117	13.1	164	6.8	yes	mother	6.8	no
8	F	2500	39	11	118	20	250	6.2	yes			nd

* liver transplantation; nd: not determined.

## Data Availability

Not applicable.

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
