# Peer review of "“Pesto” Mutation: Phenotypic and Genotypic Characteristics of Eight GCK/MODY Ligurian Patients"

_ijms, 2023, doi:10.3390/ijms24044034_

Round 1
Author Response
Response Reviewer_1
1- Clarity of language. The language of the paper needs a lot of improvement. Seeking help form a native English writer would greatly benefit its presentation
R: The language of the paper was improved, and a native English revision was done as requested.
2- Inadequate sample size. The manuscript is based on findings from a cohort of 8 patients which are too low to be statistically significant to advocate for the recognition of a novel mutation. Also, the number of male and female subjects is not similar and the clinical characteristics like age, insulin dependence, blood glucose levels, age at hyperglycemia diagnosis are so diverse to have a unifying theme. Increase the sample size to a statistically significant number would be helpful.
R: Main aim of our brief report was to describe the clinical manifestation of the insertion deletion c.1279_1358delinsTTACA, classified as likely pathogenic mutation and located in GCK exon 10. It was detected by Sanger sequencing in a small group of GCK/MODY subjects i.e. 8 affected from 6 unrelated family. In the manuscript text we aimed to underline that all of them share the origin from Liguria a small region located in the north west of Italy with a number of newborns less than 9,000 for year. Our experience is related to GCK/MODY patients genetically diagnosed and characterized from 2008 to 2022 therefore we can assume that in the last 16 years about 144,000 newborns were born in Liguria. Even GCK/MODY is the most common form of monogenic diabetes in Italy and in South of Europe, its frequency is very low in healthy population. Based on this consideration we thought that the number of patients described in this paper is not inadequate for a brief report communication. We also consider the hypothesis to enlarge our GCK MODY cases in the next few years in order to verify the frequency of this change in Ligurian population.
In the text we didn’t declare that the insertion deletion c.1279_1358delinsTTACA is novel, indeed we sentence that “It was also included in dbSNP database as rs193922274 without any frequency and it was absent from Exac, 1000G, gnomAD Exomes and gnomAD Genomes. ClinVar and ACMG classified it as “Likely pathogenic allele” and “Pathogenic” respectively. “
3 -The patients that have been enrolled for the study have no uniform characteristic features in relation to hyperglycemia, BMI or requirement of treatment. Identification of one sub-group with related clinical characteristics would make the paper more focused. This reflects a lack of understanding of the importance of clinical parameters that need to be assessed at the time of patient enrollment and should be rectified.
R: Patients were selected applying the ISPAD Clinical Practice Consensus Guidelines the diagnosis and management of monogenic diabetes in children and adolescents. In this brief report we reported 8 subjects with c.1279_1358delinsTTACA causative of GCK/MODY. We know that this is a small group and patients have no uniform characteristic features in relation to hyperglycemia, BMI or requirement of treatment; each of them has a family history of hyperglycemia. We also consider the hypothesis to enlarge our GCK MODY cases in the next few years in order to better describe these parameters as requested.
Criteria used for the selection of MODY patients to be genotyped were added in methods paragraph.
4- The discussion is too short and does not shed much light on the shortcomings or attempt to explain any of the observed characteristics in the patients. It also falls short of providing convincing justifications for the number of patients enrolled and the need for identification of “pesto” as a novel mutation.
R: In a small group of GCK/MODY Ligurian patients (N=8) we aimed to report the clinical characteristic of the indel c.1279_1358delinsTTACA. The brief report is focused mainly on two points: it could be responsible for a clinically worse form of GCK/MODY; we may postulate a possible founder effect.
Regarding to observation related to number of cases treated in the text we already reply in point 2.
Line wise comments:
Line 48: remove they before no
R: we removed they as requested
Line 49-50: MODY is a disease in itself and not a protective mechanism. The mention of the caveat of MODY not protecting against T2D seems counterintuitive and out of the context.
R: the sentence was removed from the text as requested
Line 56: Please change” nowadays” to “currently”
R: we substituted “nowadays” with “currently” as requested
Line 69: Change “glycemic” to “glycemic”
R: we corrected the word as requested

Reviewer 2 Report
Maturity Onset Diabetes of the Young (MODY) is a monogenic form of diabetes mellitus (DM). GCK/MODY is the most frequent form in Italy and it is caused by mutations of the glucokinase (GCK). Molecular analysis of the GCK coding exons was carried out by Sanger sequencing in 8 italian patients. The higher levels of HbA1c and the higher percentage of patients requiring insulin therapy compared to the previously studied Italian patients with GCK/MODY, suggest that the mutation discovered could be responsible for a clinically worse form of GCK/MODY. Moreover, based on the genotypic and clinical findings, this study investigated a critical scientific question in the subjects carrying "pesto” mutation have a more severe clinical GCK/MODY phenotype and they often require pharmacological treatment and that "pesto” mutation may have ancestral origin from Liguria. This is a well written article and the English is right with some typos. The authors only use the data of Sanger sequencing in 8 italian patients, the conclusion seems weak, more works to be done may be more convincing. In References, the latest references have fewer citations.
Author Response
Response to Reviewer_2
Maturity Onset Diabetes of the Young (MODY) is a monogenic form of diabetes mellitus (DM). GCK/MODY is the most frequent form in Italy and it is caused by mutations of the glucokinase (GCK). Molecular analysis of the GCK coding exons was carried out by Sanger sequencing in 8 italian patients. The higher levels of HbA1c and the higher percentage of patients requiring insulin therapy compared to the previously studied Italian patients with GCK/MODY, suggest that the mutation discovered could be responsible for a clinically worse form of GCK/MODY. Moreover, based on the genotypic and clinical findings, this study investigated a critical scientific question in the subjects carrying "pesto” mutation have a more severe clinical GCK/MODY phenotype and they often require pharmacological treatment and that "pesto” mutation may have ancestral origin from Liguria. This is a well written article and the English is right with some typos. The authors only use the data of Sanger sequencing in 8 italian patients, the conclusion seems weak, more works to be done may be more convincing. In References, the latest references have fewer citations.
R: Main aim of our brief report was to describe the clinical manifestation of the insertion deletion c.1279_1358delinsTTACA, classified as likely pathogenic and located in GCK exon 10. It was detected by Sanger sequencing in a small group of GCK/MODY subjects i.e. 8 affected from 6 unrelated family. In the manuscript text we aimed to underline that all of them share the origin from Liguria a small region located in the north west of Italy with a number of newborns less than 9,000 for year. Our experience is related to GCK/MODY patients genetically diagnosed and characterized from 2008 to 2022 therefore we can assume that in the last 16 years about 144,000 newborns were born in Liguria. Even GCK/MODY is the most common form of monogenic diabetes in Italy and in South of Europe, its frequency is very low in healthy population. Based on this consideration we thought that the number of patients described in this paper is not inadequate for a brief report communication. We also consider the hypothesis to enlarge our GCK MODY cases in the next few years in the following years in order to verify the frequency of this change in Ligurian population.
We verified the citation number of the reference 17 and 18 as suggested: they currently have 3 and 11 citations respectively. We agree with the opinion that reference 17 has fewer citations but to the best of our knowledge it is the only paper that describes the ancestral origin of a GCK variant in Italian population; so we consider important to include this paper as a reference.

Reviewer 3 Report
Good report
Author Response
Thank you for your revison.
Best regards
Round 2
Reviewer 1 Report
Thank you for submitting a revised version of the manuscript. After reading the revised form, I still have two major concerns that have not been completely addressed.
1. The number of patients reported is a cause of concern in my opinion. Although a reasonable explanation about the overall number of inhabitants and birth rate in the population in the reported area has been provided in the rebuttal, I would still like to see a bigger sample size for a clinical study of this nature to be able to arrive at a conclusion.
2. The overall writing and presentation of the methods and discussion can be greatly improved.
Author Response
- The number of patients reported is a cause of concern in my opinion. Although a reasonable explanation about the overall number of inhabitants and birth rate in the population in the reported area has been provided in the rebuttal, I would still like to see a bigger sample size for a clinical study of this nature to be able to arrive at a conclusion.
R: In our lab We receive requests of GCK genetic investigation from all over Italy. Unfortunately, we didn’t detect other cases resulting as carriers of the indel; consequently, it is impossible for us to increase the number of patients. We searched on pubmed “founder effect of MODY mutation” and we obtained 18 results.
In the manuscript from Delvecchio M et al “the P59S was found in 6 (27.3%) of the 22 patients, all of whom originated from a small area in Italy known as Gargano”.
The report of Dusatkova et al, declares: “In the GCK-MODY, several mutations have been shown to arise from common ancestors: a mutation which was reported in five apparently unrelated families in Oxford (p.Gly299Arg mutation) (5), a mutation in five families in Quebec (p.Val226Met).(6), three mutations in Spain (p.Val182Leu in three families from Cantabria, p.Glu399X in two families from Albacete, and p.Ala379Val in two families from Basquete) (7), a mutation in six families from Norway (p.Glu339Gly) (8), and a mutation in five families in Slovakia (c.-71G>C) (9)”.
5 Saker PJ, Hattersley AT, Barrow B et al. High prevalence of a missense mutation of the glucokinase gene in gestational diabetic patients due to a founder-effect in a local population. Diabetologia 1996: 39: 1325– 1328.
6 Henderson M, Levy E, Delvin E, Losekoot M, Lambert M. Prevalence and clinical phenotype of the p.Val226Met glucokinase gene mutation in French Canadians in Quebec, Canada. Mol Genet Metab 2007: 90: 87– 92.
7 Estalella I, Rica I, Perez de Nanclares G, et al. Mutations in GCK and HNF-1alpha explain the majority of cases with clinical diagnosis of MODY in Spain. Clin Endocrinol (Oxf) 2007: 67: 538– 546.
8 Sagen JV, Bjorkhaug L, Molnes J et al. Diagnostic screening of MODY2/GCK mutations in the Norwegian MODY Registry. Pediatr Diabetes 2008: 9: 442– 449.
9 Gasperikova D, Tribble ND, Stanik J et al. Identification of a novel beta-cell glucokinase (GCK) promoter mutation (−71G>C) that modulates GCK gene expression through loss of allele-specific Sp1 binding causing mild fasting hyperglycemia in humans. Diabetes 2009: 58: 1929– 193 k estimates regarding the age of the mutations
In our paper the number of cases (8 cases from six unrelated familes) described is in line with the cases described above (Ref 5-9). All the references above were added to the discussion paragraphs.
- The overall writing and presentation of the methods and discussion can be greatly improved.
The overall writing and presentation of the methods and discussion was improved as requested

Reviewer 2 Report
The authors explain the doubts about the small sample size that only use the data of Sanger sequencing in 8 italian patients and carefully revise the article. It will be better if they could continue to make minor revisions to the problem of fewer citations from the latest references.
Author Response
Response to Referee 2
The authors explain the doubts about the small sample size that only use the data of Sanger sequencing in 8 italian patients and carefully revise the article. It will be better if they could continue to make minor revisions to the problem of fewer citations from the latest references.
R: We revised the text including new most cited references as requested.

Round 3
Reviewer 1 Report
Dear Authors,
After reading your response to the comments, of note that similar studies have been highlighted which were conducted on a small population of patients owing to the fact that the mutant alleles are rare and including these in the discussion to lend support to the novelty and significance of your work, I believe the manuscript should be considered for publication.
Thank you!